# Stress Engineering of Magnetization Fluctuation and Noise Spectra in Low-Barrier Nanomagnets Used as Analog and Binary Stochastic Neurons

**DOI:** 10.3390/mi15091174

**Published:** 2024-09-22

**Authors:** Rahnuma Rahman, Supriyo Bandyopadhyay

**Affiliations:** Department of Electrical and Computer Engineering, Virginia Commonwealth University, Richmond, VA 23284, USA; rahmanr3@vcu.edu

**Keywords:** binary stochastic neuron, analog stochastic neuron, noise auto-correlation function, noise spectral density, telegraph noise, white noise, noise engineering, stress-modulated noise source

## Abstract

A single-domain nanomagnet, shaped like a thin elliptical disk with *small eccentricity*, has a double-well potential profile with two degenerate energy minima separated by a *small* barrier of a few kT (*k* = Boltzmann constant and *T* = absolute temperature). The two minima correspond to the magnetization pointing along the two mutually anti-parallel directions along the major axis of the ellipse. At room temperature, the magnetization fluctuates randomly between the two minima, mimicking telegraph noise. This makes the nanomagnet act as a “binary” stochastic neuron (BSN) with the neuronal state encoded in the magnetization orientation. If the nanomagnet is magnetostrictive, then the barrier can be depressed further by applying (electrically generated) uniaxial stress along the ellipse’s major axis, thereby gradually eroding the double-well shape. When the barrier almost vanishes, the magnetization begins to randomly assume any arbitrary orientation (not just along the major axis), making the nanomagnet act as an “analog” stochastic neuron (ASN). The magnetization fluctuation then begins to increasingly resemble white noise. The full width at half maximum (FWHM) of the noise auto-correlation function decreases with increasing stress, as the fluctuation gradually transforms from telegraph noise to white noise. Consistent with this trend, the noise spectral density exhibits a 1/f^*β*^ spectrum (at high frequencies) with β decreasing from 2.00 to 1.88 with increasing stress. Stress can thus not only reconfigure a BSN to an ASN, which has its own applications, but it can also perform “noise engineering”, i.e., tune the auto-correlation function and power spectral density, having applications in signal processing.

## 1. Introduction

Binary and analog stochastic neurons are powerful hardware accelerators for probabilistic computers that are adept at solving either combinatorial optimization problems in binary space [1,2,3,4,5] or temporal sequence learning/prediction in analog space [6,7]. Both types of neurons can be implemented with a single-domain low-barrier nanomagnet, such as one shaped into a thin elliptical disk with small eccentricity. The magnetization orientation encodes the neuron’s state. If the energy barrier within the nanomagnet is low compared to the thermal energy kT, but still high enough that the potential profile has the character of a *double well* with two degenerate energy minima, then the magnetization will fluctuate randomly between the two minima and the behavior will be that of a binary stochastic neuron (BSN) whose state is always either +1 or −1, albeit varying randomly in time. The probability of being in either state can be tuned by injecting a spin-polarized current into the nanomagnet [1] or by applying a voltage to induce voltage-controlled magnetic anisotropy [8]. If, on the other hand, the energy barrier is depressed enough with some external agent to erode the double-well feature, then the magnetization will be equally likely to point in any direction, i.e., the neuron’s state can assume (randomly) any value between +1 and −1, as in a true random number generator with uniform distribution. This makes it an analog stochastic neuron (ASN). The external agent thus reconfigures a BSN to an ASN.

If the nanomagnet is magnetostrictive, then the external agent can be the *electrically generated uniaxial stress of the right sign* applied along the major axis of the elliptical disk [9,10]. The sign of the stress (tensile or compressive) must be such that the sign of the product of the stress and magnetostriction is *negative* [9]. Such stress will depress the energy barrier and transform a BSN into an ASN, thereby providing a powerful route to reconfigurability in probablistic computers. The myriad uses of stress-engineered *reconfigurable* stochastic neurons have been examined in [9,10], and the modality of generating stress electrically has been described in [9] [also, see the Appendix A].

Here, we address a different topic, namely how the fluctuation/noise characteristic of the magnetization (neuronal state) changes as the internal energy barrier within the low-barrier nanomagnet (LBM) is gradually depressed with stress to reconfigure a BSN to an ASN. When no stress is applied and the energy barrier is high enough to sport a double-well appearance, the fluctuation of the magnetization mimics telegraph noise [see Figure 1]. As the energy barrier is gradually lowered, the fluctuation begins to change from telegraph noise to white noise, as shown in Figure 1. Throughout this range, the noise spectral density has a 1/f*^β^* spectrum at high frequencies. The value of β decreases with increasing stress (it will become 0 in the limit of pure white noise and 2 in the limit of pure telegraph noise). Thus, a stress-engineered low-barrier nanomagnet is not only a reconfigurable stochastic neuron but also a tunable noise source with a tunable power spectrum—a controllable nanomachine—which may have applications in communications such as noise radar technology [11,12], hardware security [13], cryptography [14] and automatic speaker classification [15].

Secure communication between devices is critical in the era of the Internet of Things (IoT). One approach is to use true random numbers for security [16]. Thermal noise and other random sources have been traditionally used to generate true random numbers but often come with excessive design complexity of circuits and post-processing to be useful for low-power and low-area applications, such as edge processing [17,18,19]. Here, we have employed a random number generator that has a miniscule footprint and very little energy consumption, making it ideal for edge applications. The noise source can be reconfigured spontaneously with a small voltage to change its spectral characteristics, making it more robust against targeted attacks.

## 2. Materials and Methods

To study the noise engineering paradigm, we simulated magnetization dynamics in a single-domain low-barrier nanomagnet at room temperature under different (barrier lowering) stresses using the Landau–Lifshitz–Gilbert–Langevin (LLGL) equation with a thermal noise term. The nanomagnet is a thin elliptical disk of cobalt with major axis 100 nm, minor axis 99 nm and thickness 5 nm. For cobalt, saturation magnetization $M_{s}$ = 10^6^ A/m, the magnetostriction coefficient $\lambda_{s}$ = −35 ppm and the Gilbert damping coefficient α = 0.01.

The coupled LLGL equations governing the temporal evolutions of the scalar components of the magnetization $m_{x}\left(t\right)$, $m_{y}\left(t\right)$ and $m_{z}\left(t\right)$—all normalized to the saturation magnetization $M_{s}$—were solved with the finite difference method [20,21]. The effect of uniaxial stress was modeled via a magnetic field term and the effect of thermal noise via another (random) magnetic field term. In all cases, the initial condition was that the magnetization was aligned close to the major axis of the nanomagnet. The time step used in the simulation was 0.1 ps.

The coupled LLGL equations describing the temporal evolution of the three components of the magnetization are
(1)$$\begin{array}{ccc} \frac{dm_{x}\left(t\right)}{dt} & = & -\gamma \left[H_{z}\left(t\right)m_{y}\left(t\right)-H_{y}\left(t\right)m_{z}\left(t\right)\right]\\ & & -\alpha \gamma \left[H_{y}\left(t\right)m_{x}\left(t\right)m_{y}\left(t\right)-H_{x}\left(t\right)m_{y}^{2}\left(t\right)-H_{x}\left(t\right)m_{z}^{2}\left(t\right)+H_{z}\left(t\right)m_{x}\left(t\right)m_{z}\left(t\right)\right]\\ \frac{dm_{y}\left(t\right)}{dt} & = & -\gamma \left[H_{x}\left(t\right)m_{z}\left(t\right)-H_{z}\left(t\right)m_{x}\left(t\right)\right]\\ & & -\alpha \gamma \left[H_{z}\left(t\right)m_{y}\left(t\right)m_{z}\left(t\right)-H_{y}\left(t\right)m_{z}^{2}\left(t\right)-H_{y}\left(t\right)m_{x}^{2}\left(t\right)+H_{x}\left(t\right)m_{x}\left(t\right)m_{y}\left(t\right)\right]\\ \frac{dm_{z}\left(t\right)}{dt} & = & -\gamma \left[H_{y}\left(t\right)m_{x}\left(t\right)-H_{x}\left(t\right)m_{y}\left(t\right)\right]\\ & & -\alpha \gamma \left[H_{x}\left(t\right)m_{z}\left(t\right)m_{x}\left(t\right)-H_{z}\left(t\right)m_{x}^{2}\left(t\right)-H_{z}\left(t\right)m_{y}^{2}\left(t\right)+H_{y}\left(t\right)m_{y}\left(t\right)m_{z}\left(t\right)\right] \end{array}$$
where α is the Gilbert damping factor of the nanomagnet material, γ is the gyromagnetic factor (a constant), $m_{i}\left(t\right)$ is the *i*-th component of the normalized magnetization at time *t*, and $H_{i}\left(t\right)$ is the *i*-th component of the effective magnetic field experienced by the nanomagnet at time *t*. The major axis of the nanomagnet is along the y-direction and the minor axis is along the x-direction.

The effective magnetic field components are given by
(2)$$\begin{array}{ccc} H_{x}\left(t\right) & = & -M_{s}N_{x}m_{x}\left(t\right)+h_{x}^{noise}\left(t\right)\\ H_{y}\left(t\right) & = & -M_{s}N_{y}m_{y}\left(t\right)+h_{y}^{noise}\left(t\right)+\frac{3}{\mu_{0}M_{s}}\lambda_{s}\sigma m_{y}\left(t\right)\\ H_{z}\left(t\right) & = & -M_{s}N_{z}m_{z}\left(t\right)+h_{z}^{noise}\left(t\right), \end{array}$$
where $N_{x}$, $N_{y}$ and $N_{z}$ are the demagnetization factors along the minor axis, major axis and out-of-plane direction (they depend on the dimensions of the major axis, minor axis and thickness), while $h_{i}^{noise}\left(t\right)=\sqrt{\frac{2\alpha kT}{\gamma \left(1+\alpha^{2}\right)\mu_{0}M_{s}\Omega \Delta t}}G_{\left(0,1\right)}^{i}\left(t\right)$, with $G_{\left(0,1\right)}^{i}\left(t\right)$ (i=x,y,z) being three uncorrelated Gaussians of zero mean and unit standard deviation, Ω being the nanomagnet volume, σ being the uniaxial stress applied along the major axis (*y*-axis) and Δt being the attempt period which is the time step of the simulation (0.1 ps).

As the magnetization fluctuates randomly owing to thermal noise, the component along the major axis $m_{y}\left(t\right)$ also fluctuates. We calculated the auto-correlation function C(τ) of $m_{y}\left(t\right)$ using the usual prescription:(3)$$C\left(\tau \right)=\int_{-\infty }^{\infty }m_{y}\left(t\right)m_{y}\left(t+\tau \right)dt.$$

We then used the Wiener–Khinchin theorem to extract the noise power spectral density spectrum S(f):(4)$$S\left(f\right)=Re\left[\int_{-\infty }^{\infty }C\left(\tau \right)e^{-2\pi f\tau }d\tau \right]=2\int_{0}^{\infty }C\left(\tau \right)cos\left(2\pi f\tau \right)d\tau .$$

## 3. Results and Discussion

The auto-correlation functions C(τ) are plotted in Figure 2 for different stress values, while the corresponding power spectral density spectra are shown in Figure 3.

At zero or low stress values, the magnetization fluctuation resembles *telegraph noise* [see Figure 1]. Telegraph noise consists of a random signal ζ(t)∈ [−1, +1] and the number of arrivals in an interval [$t_{1},t_{2}$], which is Poissonian with a distribution $\lambda (t_{1}-t_{2})$. The auto-correlation function of telegraph noise is ideally $C\left(\tau \right)=e^{-2\lambda \tau }$ [22]. In Figure 2, we plotted the auto-correlation function in both linear and log-linear scale for different values of stress [0, 2, 5 and 6 MPa]. From the latter plot, we extracted an effective Poisson parameter $\lambda_{eff}$ from the slope. These are listed in Table 1. We emphasize that the fluctuation resembles telegraph noise *only* at zero or low stress values; hence, $\lambda_{eff}$ will correspond to the Poisson parameter λ only at a low stress value and not at a higher stress value when the noise begins to deviate from the character of telegraph noise.

We also list the full width at half maximum (FWHM) of the auto-correlation functions in Table 1 for the different stress values. The energy barriers within the nanomagnet at these stress values were calculated in ref. [9].

Two features immediately stand out in the above table. First, $\lambda_{eff}$ increases with increasing stress, indicating an increase in the *arrival rate*. This is consistent with Figure 1, where we clearly see that the flips per second (or the rate of flips) is increasing with increasing stress. This happens because stress depresses the barrier, making it easier for the magnetization state to hop over the barrier. It has a very important consequence for probabilistic computing with stochastic neurons. For autonomous clockless computing, the computational speed depends on the flips per second (*fps*) [23], and increasing that quantity with stress will increase the computational speed. In the past, we have shown that the *fps* can be increased by choosing ferromagnets with low saturation magnetization [21], which, like stress, has the effect of lowering the energy barrier within the nanomagnet. Here, we show that this objective can be also achieved by applying stress. The former approach is not *reconfigurable* since the *fps* could not be changed once the nanomagnet (with a particular saturation magnetization) was fabricated; however, here, one can change the *fps* at will with electrically generated stress.

The second feature to note in Figure 2 is that the FWHM decreases with increasing stress and the auto-correlation function gradually approaches a δ-function as the stress is increased. The auto-correlation function of white noise is a δ-function. Hence, as we increase stress and lower the energy barrier within the nanomagnet, the magnetization fluctuation gradually transforms from telegraph noise toward white noise.

We also point out two features in the spectral density plots shown in Figure 3. First, it is evident from the spectra that the noise begins to develop higher-frequency components with increasing stress. This is also a manifestation of the fact that the flips per second are increasing with stress (which lowers the energy barrier within the nanomagnet), *and* this is also a telltale sign of the noise gradually transforming from telegraph toward white. The other feature is less obvious. We calculated the integrated noise power $\int_{0}^{\infty }S\left(f\right)df$ at various values of stress and tabulated them in Table 2. Within numerical inaccuracies, this quantity is invariant under stress. This is expected. Stress does not inject or extract any power from the system; hence, the integrated noise should be relatively independent of stress.

Finally, we note that the power spectral density of ideal telegraph noise has the form of a Cauchy density function [22] and hence will have the form
(5)$$S\left(f\right)=\frac{\lambda }{\pi f^{2}+\lambda^{2}},$$
which, in the high frequency limit, becomes approximately
(6)$$S\left(f\right)\approx \frac{\lambda }{\pi }\left[\frac{1}{f^{2}}\right].$$

Because the noise has a telegraph character at zero or low stress values and increasingly acquires the characteristic of white noise at high stress values, we expect the noise spectra at high frequencies to exhibit a 1/$f^{\beta }$ dependence, where β will be close to 2 at a zero stress value and decrease at higher stress values (for white noise, β = 0).

We fitted the noise power spectra at high frequencies with 1/$f^{\beta }$ and found that the β values vary from 2 (no stress) to 1.88 (6 MPa stress). The fitting is shown in the Appendix. The β value does decrease with stress, but never quite approaches zero, which would be characteristic of white noise. Thus, the noise remains primarily as telegraph noise within the stress values considered here, but begins to develop the white noise characteristic as the stress is increased to lower the energy barrier within the nanomagnet.

## 4. Conclusions

Low-barrier nanomagnets shaped like thin elliptical disks with low eccentricity are popular hardware accelerators for stochastic neurons, where the randomly fluctuating magnetization orientation encodes the neuronal state. Here, we have shown that this construct has another application, namely an *engineered noise source*. The magnetization fluctuation at room temperature can be made to transform from telegraph noise toward white noise by applying uniaxial stress of the right sign along the major axis. This changes the auto-correlation function and the noise power spectral density, with potential applications in communication engineering.

## Figures and Tables

**Figure 1 micromachines-15-01174-f001:**
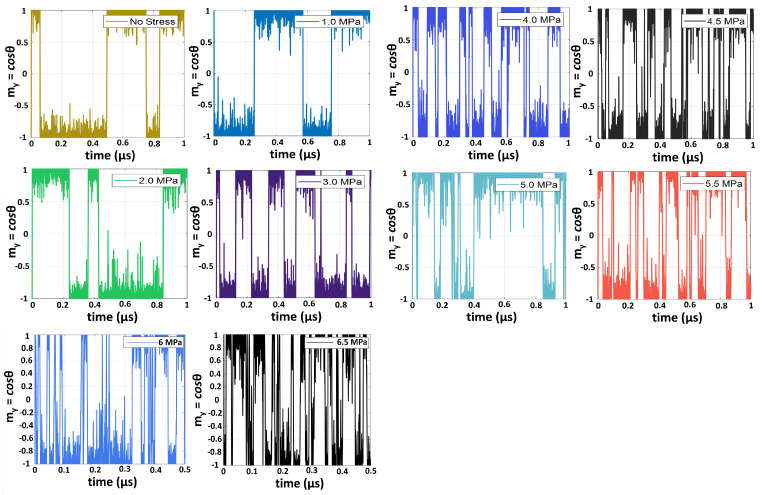
Temporal fluctuations in the magnetization component along the major axis of a Co nanomagnet shaped like an elliptical disk, with major axis = 100 nm, minor axis = 99 nm and thickness = 5 nm. The fluctuations are shown at different values of uniaxial tensile stress applied along the major axis. Note that the noise gradually transitions from telegraph to nearly white with increasing stress, which increasingly depresses the energy barrier within the nanomagnet. Reproduced from [9] with permission from the Institute of Physics.

**Figure 2 micromachines-15-01174-f002:**
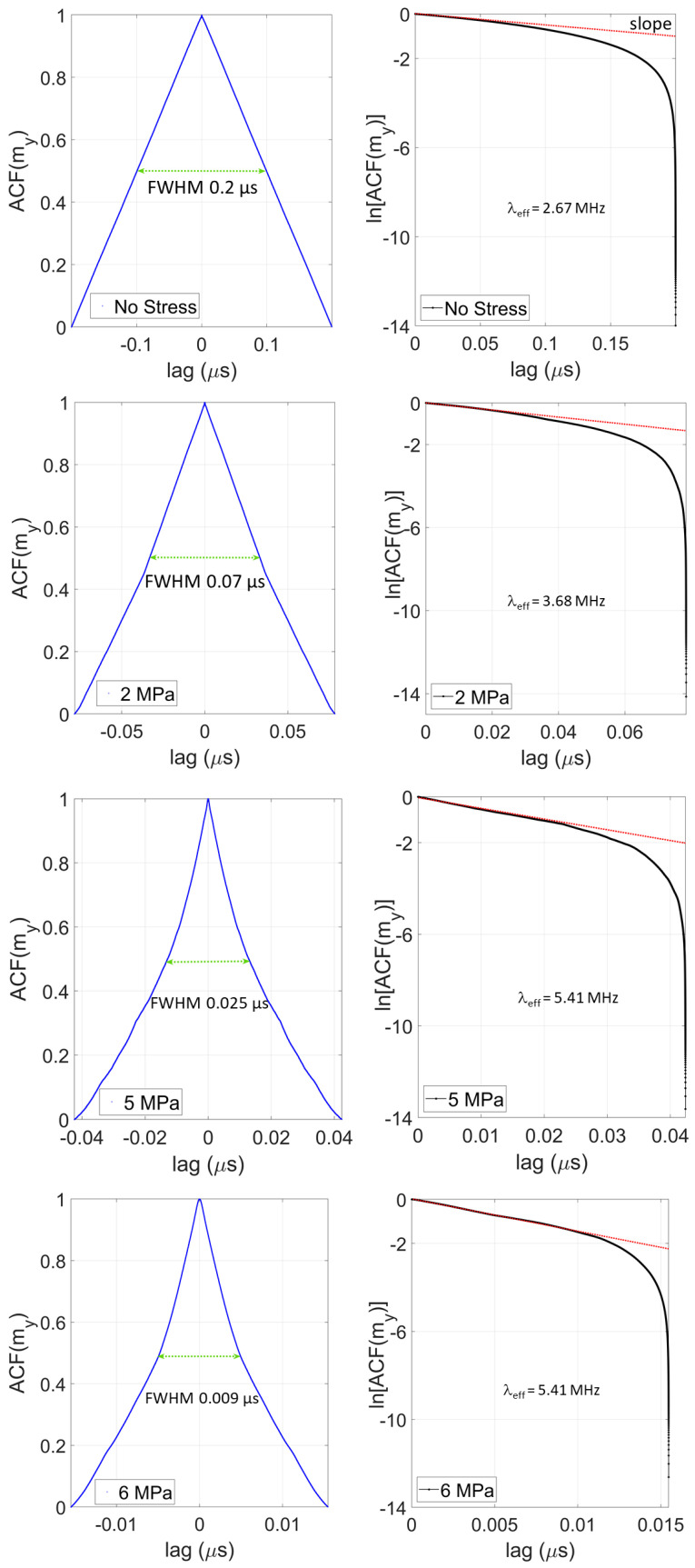
Calculated auto-correlation functions of the fluctuations in the magnetization component along the major axis of the nanomagnet at a temperature of 300 K for different values [0, 2, 5 and 6 MPa] of the uniaxial tensile stress (applied along the major axis of the nanomagnet). The plots are shown in both linear and log-linear scales. Also shown are the full width at half maximum (FWHM) values of the auto-correlation functions and the parameter $\lambda_{eff}$ at different stress values.

**Figure 3 micromachines-15-01174-f003:**
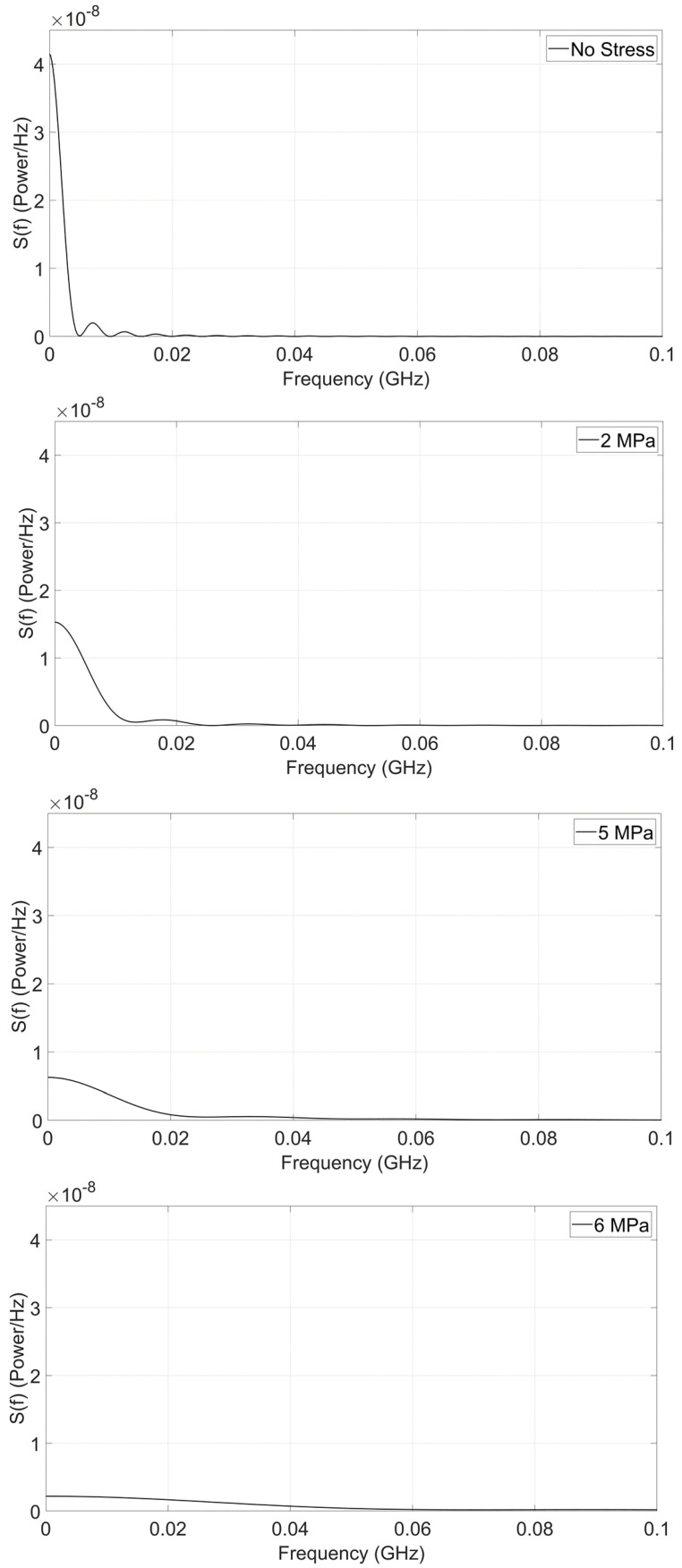
Calculated noise power spectral density of the fluctuations in the magnetization component along the major axis of the nanomagnet at a temperature of 300 K for different values [0, 2, 5 and 6 MPa] of the uniaxial tensile stress (applied along the major axis of the nanomagnet). The power is expressed in arbitrary units.

**Table 1 micromachines-15-01174-t001:** The parameter $\lambda_{eff}$ and the FWHM of the auto-correlation function at different stress values.

Stress (MPa)	$\lambda_{\mathrm{eff}}$ (MHz)	FWHM (μs)
0	2.67	0.200
2	3.68	0.070
5	5.41	0.025
6	5.41	0.009

**Table 2 micromachines-15-01174-t002:** Integrated noise power $\int_{0}^{\infty }S\left(f\right)df$ at different stress values.

Stress (MPa)	$\int_{0}^{\infty }S\left(f\right)\mathrm{df}$ (Arb. Units)
0	0.2090
2	0.2149
5	0.1983
6	0.1836

## Data Availability

No new data were created or analyzed in this study. Data sharing is not applicable to this article.

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
