# Peer review of "Stress Engineering of Magnetization Fluctuation and Noise Spectra in Low-Barrier Nanomagnets Used as Analog and Binary Stochastic Neurons"

_micromachines, 2024, doi:10.3390/mi15091174_

Round 1

Reviewer 1 Report

Comments and Suggestions for Authors

This work about a theoretical study of a metallic nanostructure seems interesting in its proposal, but it lacks of the proper profoundness and coverage of the potential phenomena occurring in terms of dimensions and quantum effects. For instance, it focus only on one set of dimensions, and the results are somewhat vague since there is not a base reference point on this. More importantly, how are the authors dealing with quantum effects that certainly arise at this scale? these effects would strongly affect the magnetic properties of the material that are normally observed at the macroscale. Also, I think the amount of results is not enough for a research paper. It seems to me that the two figures with multiple panels and two tables which describe the results can be easily summarized (and it would be good, since it would help to contrast the results) in two simple figures and only one table.

In summary, this work could be OK in average quality terms, but not enough in its extension and relevance in its current state.

Author Response

Comment: For instance, it focus only on one set of dimensions, and the results are somewhat vague since there is not a base reference point on this.

Response:  The dimensions are not important as long as the nanomagnet is single domain, which it certainly is for the chosen dimensions. If we change the dimensions, the shape anisotropy barrier will change. Nothing else will change.  What reference point is the reviewer talking about? This is not clear.

Comment: More importantly, how are the authors dealing with quantum effects that certainly arise at this scale? these effects would strongly affect the magnetic properties of the material that are normally observed at the macroscale. 

Response: This is nanomagnetism, not semiconductor quantum dot physics. We are dealing with magnons, not electrons since we are looking at magnetic properties, not electronic properties. There is nothing like  phase coherence here to induce "quantum effects". Nanoscale has other effects in magnetism, but not what this reviewer seems to be alluding to. This comment tells us that this reviewer is not familiar with this field. This field is very different from semiconductor nanophysics.

Comment: Also, I think the amount of results is not enough for a research paper. It seems to me that the two figures with multiple panels and two tables which describe the results can be easily summarized (and it would be good, since it would help to contrast the results) in two simple figures and only one table.

Response: We disagree. We need multiple results to show the evolution of the noise characteristics with stress. It is not possible to show the evolution with just two figures. Low barrier nanomagnets are used for probabilistic computing and there is vast literature on this field, some of which is cited in our bibliography. The evolution of noise with an external agent is important for this type of computing since it has implications for computing speed and convergence. The ability to manipulate noise is a new and vital capability in the world of probabilistic computing, which this reviewer did not grasp because he/she obviously does not have a background in this field. The capability we showcase is also useful for noise engineering in communication channels, stealthy radars, etc. 

It is obvious that this reviewer is not from this field. He/she should have declined this review assignment since it is outside his/her domain of expertise.

Reviewer 2 Report

Comments and Suggestions for Authors

Some suggestions to improve the quality of the manuscript:

1. In Fig. 2, change or define the abbreviation ACF. Maybe using the C(tau) description in the graphs is better. Define lag(microsec) parameter.

2. Check the reference list and correct references 12, 13, and 14. Also, check references 17 and 20, which are the same.

3. It might be better to add a figure to support the findings presented in the last paragraph of section 3 (lines 142 to 151).

Author Response

Comment 1: 1. In Fig. 2, change or define the abbreviation ACF. Maybe using the C(tau) description in the graphs is better. Define lag(microsec) parameter.

Response 1: We agree. This has been done in the revised manuscript.

Comment 2: Check the reference list and correct references 12, 13, and 14. Also, check references 17 and 20, which are the same.

Response 2: Reference 20 has been eliminated since it is the same as ref. 17. Thank you for catching this. The author name has been corrected in ref. 13.

Comment 3: It might be better to add a figure to support the findings presented in the last paragraph of section 3 (lines 142 to 151).

Response: We agree. A figure has been added in the appendix.

We thank the reviewer for the helpful comments.

Reviewer 3 Report

Comments and Suggestions for Authors

This work touched on a very interesting but overlooked point in strain-controlled magnetism, namely, the different noise characteristics and their influence to the internal energy barrier (re)-shaping in the nanomagnets.

Rigorous theoretical derivation was present and the conclusion is supported by the data. 

I believe the paper is worth publishing in Micromachines. 

A minor comment, the stress was represented in generic unit of MPa, but it would be helpful to make direct connections to real-world materials so that experimentalist can gain a rough idea, e.g., how much electric voltage (and other external controls) would be relevant in measuring the effect outlined in the theoretical scheme, using one of two standard materials and their parameters. 

Author Response

We thank the reviewer for a very insightful comment. 

Comment 1: A minor comment, the stress was represented in generic unit of MPa, but it would be helpful to make direct connections to real-world materials so that experimentalist can gain a rough idea, e.g., how much electric voltage (and other external controls) would be relevant in measuring the effect outlined in the theoretical scheme, using one of two standard materials and their parameters. 

Response: We completely agree and have introduced an Appendix to address this matter.